# Defining the sediment prokaryotic communities of the Indian River Lagoon, FL, USA, an Estuary of National Significance

**David J. Bradshaw, II**[1]*, **Nicholas J. Dickens**[1], **John H. Trefry**[2], **Peter J. McCarthy**[1]

**1** Department of Biological Sciences, Harbor Branch Oceanographic Institute at Florida Atlantic University, Fort Pierce, FL, United States of America, **2** Department of Ocean Engineering and Marine Sciences, Florida Institute of Technology, Melbourne, FL, United States of America

* dbradshaw2015@fau.edu

**Data Availability Statement:** Raw sequences have been uploaded into the National Center for Biotechnology Information Sequence Read Archive (PRJNA594146) (https://www.ncbi.nlm.nih.gov/

## Abstract

The Indian River Lagoon, located on the east coast of Florida, USA, is an Estuary of National Significance and an important economic and ecological resource. The Indian River Lagoon faces several environmental pressures, including freshwater discharges through the St. Lucie Estuary; accumulation of anoxic, fine-grained, organic-rich sediment; and metal contamination from agriculture and marinas. Although the Indian River Lagoon has been well-studied, little is known about its microbial communities; thus, a two-year 16S amplicon sequencing study was conducted to assess the spatiotemporal changes of the sediment bacterial and archaeal groups. In general, the Indian River Lagoon exhibited a pro-karyotic community that was consistent with other estuarine studies. Statistically different communities were found between the Indian River Lagoon and St. Lucie Estuary due to changes in porewater salinity causing microbes that require salts for growth to be higher in the Indian River Lagoon. The St. Lucie Estuary exhibited more obvious prokaryotic seasonality, such as a higher relative abundance of Betaproteobacteriales in wet season and a higher relative abundance of Flavobacteriales in dry season samples. Distance-based linear models revealed these communities were more affected by changes in total organic matter and copper than changes in temperature. Anaerobic prokaryotes, such as Campylobacter-ales, were more associated with high total organic matter and copper samples while aerobic prokaryotes, such as Nitrosopumilales, were more associated with low total organic matter and copper samples. This initial study fills the knowledge gap on the Indian River Lagoon bacterial and archaeal communities and serves as important data for future studies to compare to determine possible future changes due to human impacts or environmental changes.

## Introduction

The Indian River Lagoon (IRL) is an Estuary of National Significance located on Florida's east coast (USA) [1]. The lagoon has a total estimated annual economic value of $7.6 billion [2]. It

bioproject/PRJNA594146). The Snakemake file used for QIIME2 analysis and any subsequent scripts used in statistical analysis can be found on GitHub (https://github.com/djbradshaw2/General_16S_Amplicon_Sequencing_Analysis).

**Funding:** This research was supported by the following organizations and agencies: the Harbor Branch Oceanographic Institute (HBOI) Foundation (https://hboifoundation.org/) (DJB, NJD), the Everglades Foundation (https://evergladesfoundation.org/) (DJB), and the Save Our Seas Specialty License Plate which are granted through the HBOI Foundation (https://myfloridaspecialtyplate.com/save-our-seas.html) (PJM).The funders had no role in study design, data collection and analysis, decision to publish, or preparation of the manuscript.

is connected, at its southern end, to the St. Lucie Estuary (SLE), another important resource for the area [3]. The IRL has a high biodiversity because it is located at the border between temperate and sub-tropical regions, allowing it to have plant and animal species from both climates [4]. The IRL faces similar environmental issues to other estuaries, including freshwater inputs, eutrophication, organic matter, and metal contamination [1, 5].

Freshwater is introduced into the IRL via runoff from local waterways and, most prominently, discharges from Lake Okeechobee, which are diverted into the SLE through the C-44 canal during periodic releases based upon the Lake's water level [3]. This introduction of freshwater and its associated contaminants causes problems for the ecosystem [3]. The high flow conditions and drastic reduction of salinity associated with these discharges is so prominent in the SLE that it has reduced and prevented the recovery of oyster reefs [6]. Freshwater discharges also introduce dissolved organic material and plant matter that settles into the sediment to become the fine grained, highly organic sediment known as "IRL muck" [7]. "IRL muck" (hereinafter referred to as muck) is defined as sediment that has at least 75% water content, and the remaining solids fraction has at least 60% fines and 10% total organic matter (TOM) [8]. Muck can lead to various negative ecological impacts including nutrient flux in the water column triggering algal blooms and turbidity which damages seagrasses by blocking sunlight [9]. About 10% of the IRL is covered in muck ranging in depths from centimeters to several meters [1, 8, 10]. Anoxia is associated with muck and can also be caused by freshwater discharges carrying contaminants from agriculture and urban development [8, 11, 12]. A shift to an anoxic state alters the most energetically favorable terminal electron acceptors for microbes, altering their population, diversity, and functions [11, 12]. A study of Chesapeake Bay (MD, USA), compared the prokaryotic communities of anoxic, organic-rich, silty-clay sediments to organic-poor, sandy sediments and found major differences due to the former having bacterial members that contribute to the high sulfide and methanogenic conditions in the area [13].

Other pressures on the IRL include contaminants such as trace metals which can be transported in discharges and runoff as part of metal-dissolved organic matter complexes that precipitate onto the sediment once this freshwater meets the brackish water of the IRL [7, 8, 14]. A survey of the northern IRL found several sites with metals above normal levels, while a survey in the SLE found a large accumulation of phosphorus and Cu, the latter likely due to Cu-containing fungicides or cuprous oxide anti-fouling paints used in marinas [15–18]. The interaction of microbes with heavy metals affects their chemical forms and therefore their solubility, mobility, bioavailability, and toxicity [19]. In turn, prokaryotic assemblages can be altered by the presence of heavy metals, which can lead to a decrease in microbial diversity and functional redundancy [5, 20–23]. A study in Chile, found that there was a significant decrease in the abundance of bacteria in copper contaminated sites, while the abundance of archaea was similar to a less contaminated site, likely due to copper resistance mechanisms [24].

The true extent of the IRL's biodiversity cannot be understood without information on its microbial communities [25]. Sediment microbes, especially in estuaries, face a wide range of physicochemical gradients that can cause shifts in the microbial taxonomy as well as microbial functional capabilities [14, 26, 27]. This study was carried out to provide the first data on the bacterial and archaeal communities present in the IRL and to explore potential diversity changes due to differences between: the IRL and SLE due to their different magnitudes of freshwater discharges; samples with zero muck characteristics (sandy or 0MC) and three muck characteristics (muck or 3MC); and non-Cu contaminated and Cu-contaminated, high TOM samples.

## Materials and methods

### Site selection

Fifteen sites were chosen based upon either being adjacent to continuous water quality monitoring stations (11 sites) [28, 29] or known to be muck [Manatee Pocket (MP) and Harbor Branch Channel (HB)] or sandy [Jupiter Narrows (JN) and Hobe Sound] sites (Fig 1 and S1 Table). During the second year of sampling, two additional marina sites [Harbortown Marina (HT) and Vero Beach Marina] and two nearby less impacted sites (Barber Bridge and Round Island) were added. The purpose of choosing these sites across a large area of the IRL and SLE

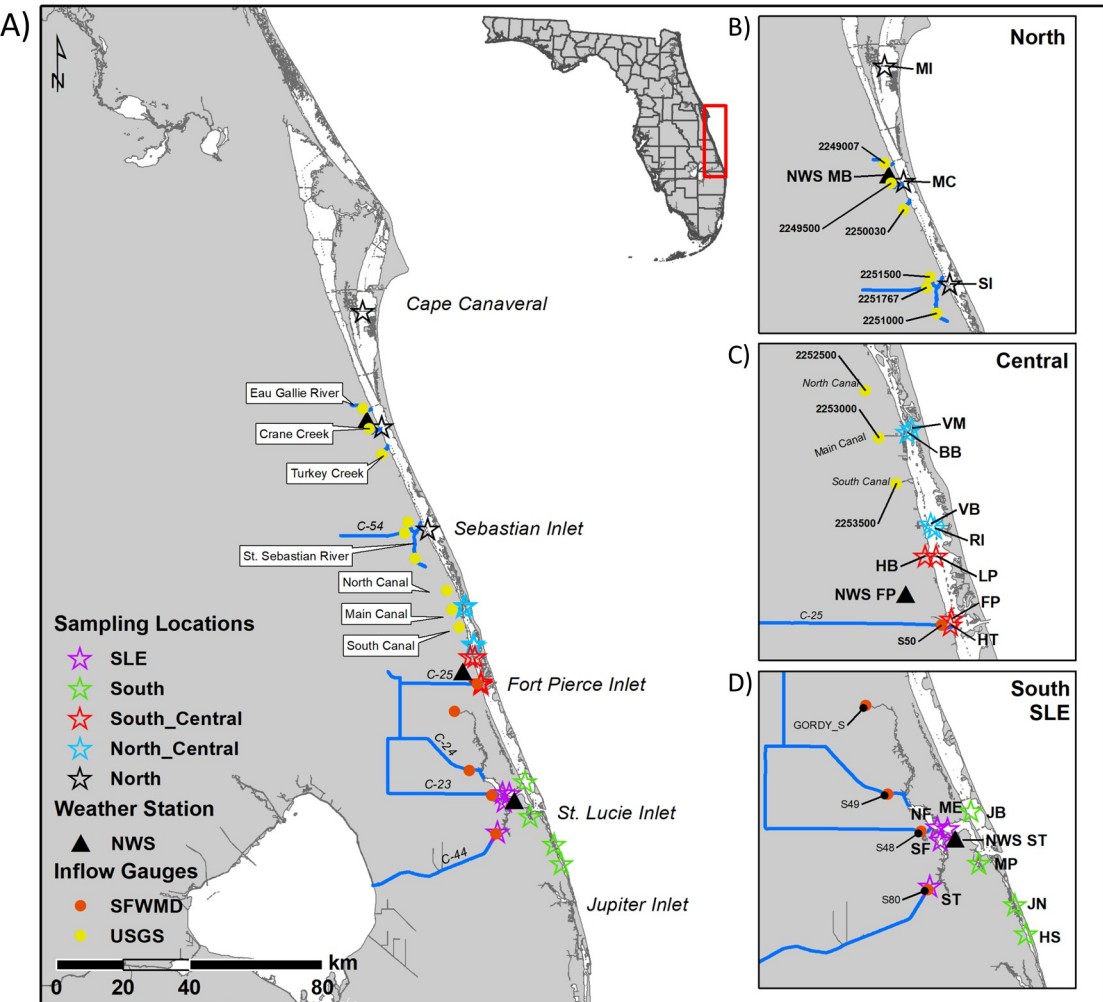

**Fig 1. Sampling area map.** The sampling area with stars indicating the site locations, triangles the location of the National Weather Service (NWS) [30] monitoring stations in Melbourne (MB), Fort Pierce (FP), and Stuart (ST), and circles the inflow gauge locations [United States Geological Service (USGS) [32] in yellow and the South Florida Water Management District (SFWMD) [31] in red]; the streams or canals associated with these gauges are denoted by blue lines. In Maps A and B black stars are the North Indian River Lagoon (IRL) sites: Merritt Island Causeway (MI), Melbourne Causeway (MC) and the Sebastian Inlet (SI). Blue stars (maps A and C) are the North Central IRL sites: Vero Beach Marina (VM), Barber Bridge (BB), Vero Beach (VB), and Round Island (RI). Red stars (maps A and C) are the South Central IRL sites: Harbor Branch Channel (HB), Linkport (LP), Fort Pierce (FP), and Harbortown Marina (HM). Green stars (maps A and D) are the South IRL sites: Jensen Beach (JB), Manatee Pocket (MP), Jupiter Narrows (JN) and Hobe Sound (HS). Maroon stars (maps A and D) are the St. Lucie Estuary (SLE) sites: North Fork (NF), South Fork (SF), Middle Estuary (ME), and South Fork 2 (ST). This map was created by using a map image from the publicly available Florida Fish and Wildlife Conservation Commission using ArcGIS [33]. GPS coordinates for all study sites and environmental monitoring stations are located in S1 Table.

was to gather samples of different sample types (0MC, 3MC, Cu-contaminated muck, non-Cu contaminated muck) across different regions to focus on the changes due to sediment characteristics that were similar across samples. Using information from NOWData (National Weather Service), the average monthly temperature and average monthly rain sum from years 1990–2018 was obtained for the Melbourne Area, Fort Pierce Area, and Stuart 4 E Stations [30]. Streamflow data was taken from DBHYDRO (South Florida Water Management District) and from the United States Geological Services website [31, 32].

## Sample collection

A total of 204 sediment samples were taken during four sampling periods over a two-year period, with two sets of samples during the wet (May-Oct) and dry seasons (Nov-Apr). No permits were required for the described study, which complied with all relevant regulations. Samples were taken in triplicate at each site to get representative samples from fifteen sites during the first two sampling periods, and nineteen sites during the latter two sampling periods. Representative samples were taken during the following months: 45 total samples during Aug-Sep 2016 (W16), 45 during Mar-Apr 2017 (D17), 57 during Oct-Nov 2017(W17), and 57 during Apr 2018 (D18). Samples were collected with published methods [8, 22]. A Wildco® Ekman stainless steel bottom grab sampler was deployed from a boat to collect sediment in triplicate. The first replicate's sediment temperature was determined with a thermometer. The top 2 cm of each replicate was subsampled with an ethanol-sterilized plastic spoon. Three sub-samples of each replicate were taken to assess the prokaryotic community (A), dry sediment characteristics (B), and wet sediment characteristics (C). They were sealed and placed on ice until returned to the lab. Subsamples A and B were collected in sterile 50-mL Falcon tubes and 75-mL polystyrene snap-cap vials, respectively, and stored at -20˚ C. The remnants of the top 2 cm of the sediment were placed in a double-bagged Ziploc freezer bag as Subsample C and stored at 4˚ C. When ready for analysis, subsample B was thawed and dried for 48 hours at 60˚C on pre-weighed, acid-washed, glass petri dishes. The differences in pre- vs post-drying weights were used to determine water content. The dried sediment was broken up with an acid-washed mortar and pestle and sieved to remove the fraction above 2 mm (coarse) from the sample. The remaining sand (2 mm-0.063 mm) and fines (<0.063 mm) fractions were kept for further analysis. All collection plasticware was soaked in 5% $HNO_3$ for a minimum of 24 hours and then rinsed three times in ultrapure water (18.2 MΩ).

## Metal analysis

Triplicate samples at each site were analyzed for heavy metals. Acid digests were prepared by modification of methods described in three studies [8, 34, 35]. Briefly, 1 g (+/- 0.0003g) of dried and sieved Subsample B was digested in 1 M HCl for one hour at 30˚ C with shaking at 150 rpm. Digests were filtered with DigiTubes® and DigiFilters® (0.45 μm) (SCP Sciences, Champlain, NY). Cu and Fe were measured with a four-point calibration curve on a Perkin Elmer 4000 atomic absorption spectrometer (Perkin Elmer, Waltham, MA). The calibration curve was rechecked every ten to twelve samples to account for absorbance drift. Reagent samples (2% $HNO_3$, 1 M HCl, ultrapure water (18.2 MΩ)) and a method control sample were analyzed to check for contamination.

## Sediment physical characteristics

A modified procedure was used to determine sediment characteristics [8]. One gram of dried and sieved Subsample B sediment was heated in a 550˚ C muffle furnace for four hours to burn off the organic matter. The sediment weight loss was calculated and reported as percent

TOM. Grain size was determined by wet sieving 10–30 g of Sample C and drying to constant weight. The gravel, sand, and fines percentages of the total dry weight were determined. PWS was determined by centrifuging 20 g of Sample C and measuring the salinity of the resulting liquid with a portable refractometer.

## Sequence analysis

DNA was extracted from 0.25–0.3 g of sediment with the Qiagen PowerSoil DNA Isolation Kits (Hilden, Germany), and its quality checked with a Nanodrop 2000 (Oxford Technologies, Oxford, UK). Samples were sent to Research and Testing (Lubbock, TX, USA) for MiSeq 16S sequencing to amplify the bacterial/archaeal 16S V4 region with the modified primers used by the Earth Microbiome Project of 515F (GTG**Y**CAGCMGCCGCGGTAA) and 806R (GGACTAC**N** VGGGTWTCTAAT) [36, 37]. The raw sequences were trimmed to remove the primers and quality-filtered with the FastX and TrimGalore programs respectively [38, 39]. Quantitative Insights Into Microbial Ecology 2's (QIIME2) (version 2018.11) [40] standard 16S workflow was used for analysis, and a Snakemake file was used for the orchestration for reproducibility [41, 42]. Sequences were joined with VSEARCH [43]. Next, they were denoised with Deblur [44] run with default parameters, with the exceptions of the minimum reads parameter set to 0 to account for metadata categories with smaller sample sizes and trim length set to 232 bases. Amplicon Sequence Variants (ASVs) were annotated with a Naïve-Bayes classifier based on the scikit-learn system and the SILVA database [45, 46] (version 132). Mitochondrial, chloroplast and unassigned sequences were filtered from the samples. The ASVs were aligned with MAFFT [47] and then masked [40, 48] to make a phylogenetic tree with FASTTREE [49] that was then midpoint-rooted. Raw sequences have been uploaded into the National Center for Biotechnology Information Sequence Read Archive (PRJNA594146) [50].

## Statistical analysis

RStudio [51, 52] (R Version 3.6.1) was used for data manipulation, visualization, generation of alpha diversity statistics (Shannon), and data manipulation. Analyses were run with the following library versions: phyloseq (1.28.0) [53], vegan (2.5–5) [54], ggplot2 (2_3.2.0) [55], reshape (0.8.8) [56], tidyverse (1.2.1) [57], and FSA (0.8.25) [58]. ASVs that did not have at least ten sequences associated with them across all samples were removed [40, 59]. Although samples were taken in triplicate at each site, the samples were not pooled for each site during each sampling period due to differences in measured environmental variables even at the site level.

PRIMER7/PERMANOVA+ was also used to analyze the data [60–62]. The environmental data was checked for highly colinear variables, greater than 0.70 [63], by generating draftsman plots. This showed that TOM was positively colinear with water content, percent fines, and Fe, and negatively colinear with percent sand. This allowed TOM to represent all these variables in future analyses. The remaining environmental variables were normalized. The biological data was square root transformed, then used to make a Bray-Curtis dissimilarity matrix to create principal coordinates of analyses. Distance-based linear models were made with a stepwise selection procedure, an AICc (An Information Criterion) selection criteria, 9 999 permutations, marginal tests, and a distance-based redundancy analysis plot [22]. Overall and pairwise permutational analysis of variance were conducted with 9 999 permutations, the unrestricted method, Type III Sum of Squares, and Monte Carlo p-values [5, 64]. Overall statistical significance of environmental data and alpha diversity metrics were determined with Kruskal-Wallis or Mann Whitney U tests for categories with greater than two or just two subcategories, respectively [65, 66]. Pairwise testing was conducted with the Dunn method [67]. All reported p-values were considered statistically significant if less than 0.05 after multiple testing

correction with the Benjamini-Hochberg (BH) method [68]. The Snakemake file used for QIIME2 analysis and any subsequent scripts used in statistical analysis can be found on Github (https://github.com/djbradshaw2/General_16S_Amplicon_Sequencing_Analysis) [69]. Measured environmental data and metadata can be found in S2 and S3 Tables, respectively.

# Results

## Weather and streamflow discharges

Measured air temperatures were higher during each of the sampling periods than historical temperatures (1990–2018), with W16 being the hottest, followed by W17, D18, and D17 (S1 Fig). Every sampling period besides D17, which was drier than usual, was wetter than usual especially W17, for which rainfall more than doubled.

On average, the stream/canal that had the largest average monthly discharge during the entire survey period (06/2016 to 06/2018) was the C44 canal (792 ft$^3$/s) and the lowest was the Eau Gallie River (15.46 ft$^3$/s) (S4A Table). The mean discharge across all streams/canals was greatest during Sept-Nov 2017 (616 ft$^3$/s), which was 3.1x higher than Jul-Sept 2016 (197 ft$^3$/s), 28x higher than Mar/Apr 2018 (22 ft$^3$/s) and 36x higher than Feb/Mar/Apr 2017 (17 ft$^3$/s) (S4B Table). Four of these streams/canals are direct tributaries of the SLE and they had an average monthly discharge (363 ft$^3$/s) that was 3.3x higher than the ten IRL streams/canals (110 ft$^3$/s) during the entire survey period (S4C Table).

## Porewater salinity and sediment temperature

Porewater salinity (PWS) and sediment temperature were measured to assess changes between sampling periods (Fig 2). Dunn testing indicated that IRL W16 and W17 sampling periods were significantly different (BH p-values < 0.05) from each other as well as both the D17 and D18 periods, although these two were significantly similar (BH p-value = 0.16) (Fig 2A and S5 Table). In the SLE, W16 and W17 were significantly similar to one another (0.65) but different from D17 and D18, which were also statistically similar to one another (0.23). The highest mean sediment temperature occurred during the W16, and Dunn testing revealed that all sampling periods were significantly different from one another except for the D17 and W17 temperatures for both the IRL (0.60) and SLE (0.052).

PWS generally increased towards the southern IRL while the SLE had the highest interquartile range, but the lowest mean (Fig 2B). All sections of the IRL were significantly different from the SLE, while only the North IRL sites were found to be statistically lower than the South IRL sites (BH p-value = 0.045) (Fig 2B and S5 Table). Sediment temperature did not vary greatly across locations, ranging from the highest mean of 26.5˚C (South) to the lowest of 24.4˚C (South Central). Each of the Location subcategories were not statistically different from one another, except for the South Central IRL being significantly lower than the South (0.00037).

## Muck and copper

The sites that, on average, exceeded three muck characteristics were Middle Estuary and South Fork, while those that exceeded at least one of the thresholds were HB, HT, Melbourne Causeway (MC), and MP (Fig 3). None of the other 13 sites exceeded the thresholds on average. Out of the 204 samples, 40 were considered muck since their sediment characteristics exceeded three thresholds (3MC), 14 only exceeded two thresholds, 10 samples exceeded one, and 140 exceeded none (0MC) (S3 Table). 3MC samples (water content = 81%, TOM = 24%, and silt/clay percentage = 81%) had 2.4x higher water content, 7.2x higher TOM, and 9.6x higher silt/clay percentages on average than 0MC samples (3.3%, 34%, 8.4%).

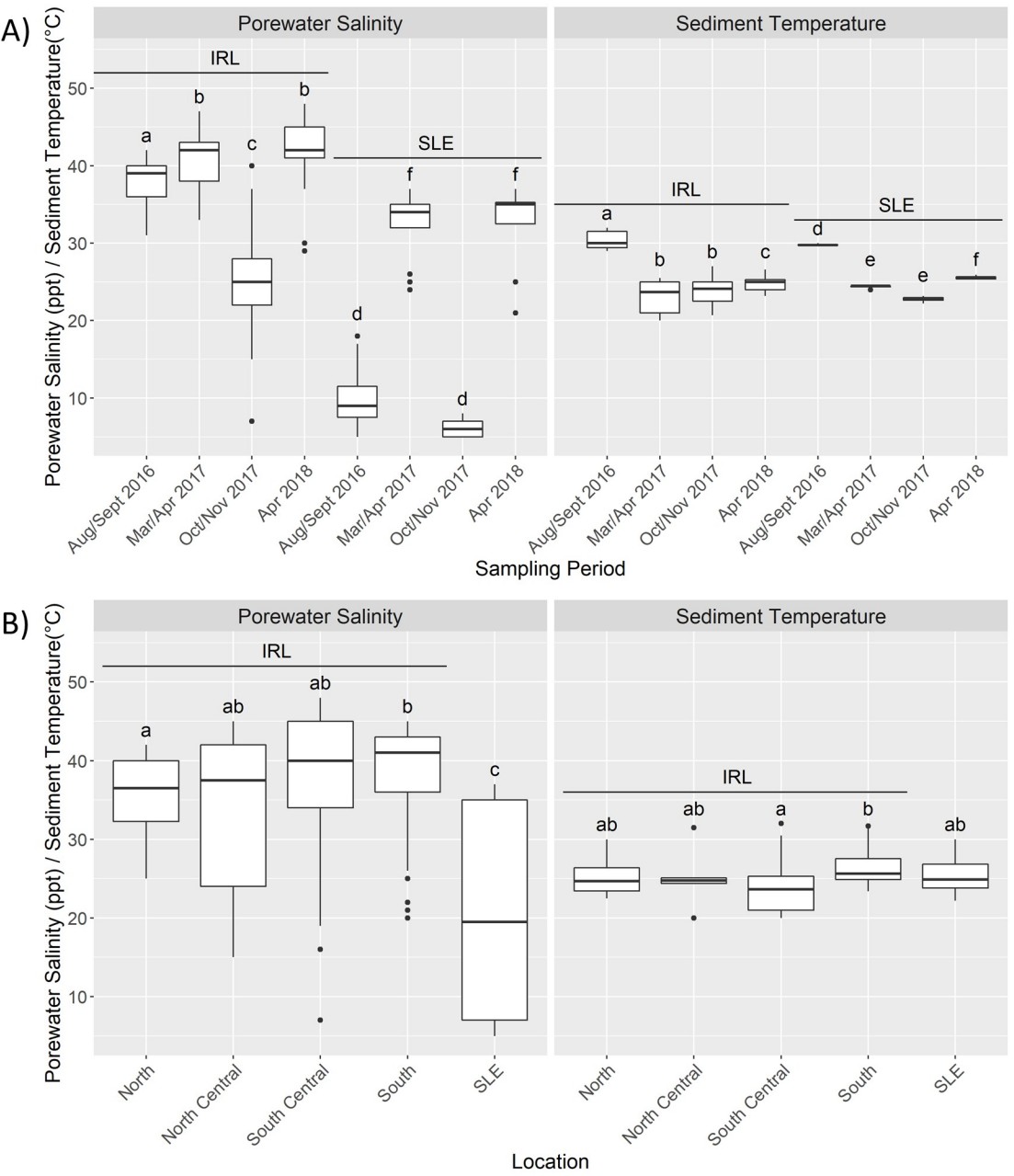

**Fig 2. Porewater salinity and sediment temperature patterns.** Porewater salinity (PWS) (left) and sediment temperature (right) by Estuary by Sampling Period (A) and by Location (B) for the Indian River Lagoon (IRL) and St. Lucie Estuary (SLE). Bars denote largest and smallest values within 1.5*the interquartile range, middle line is the median, ends of boxes are the first and third quartiles. The letters on top of each boxplot denote the results from the pairwise Dunn test with different letters denoting statistical significance (Benjamini-Hochberg adjusted p values < 0.05). In A the letters show how each of the sampling periods were different within each estuary but do not denote inter-estuary comparisons.

A sample was considered to have high TOM if it exceeded 10% and high Cu if it exceeded 65 µg/g [8, 70]. Most samples that exceed at least one of the muck characteristics also had high TOM (62/64) (Fig 4 and S3 Table). The sites that had samples with both high TOM and high

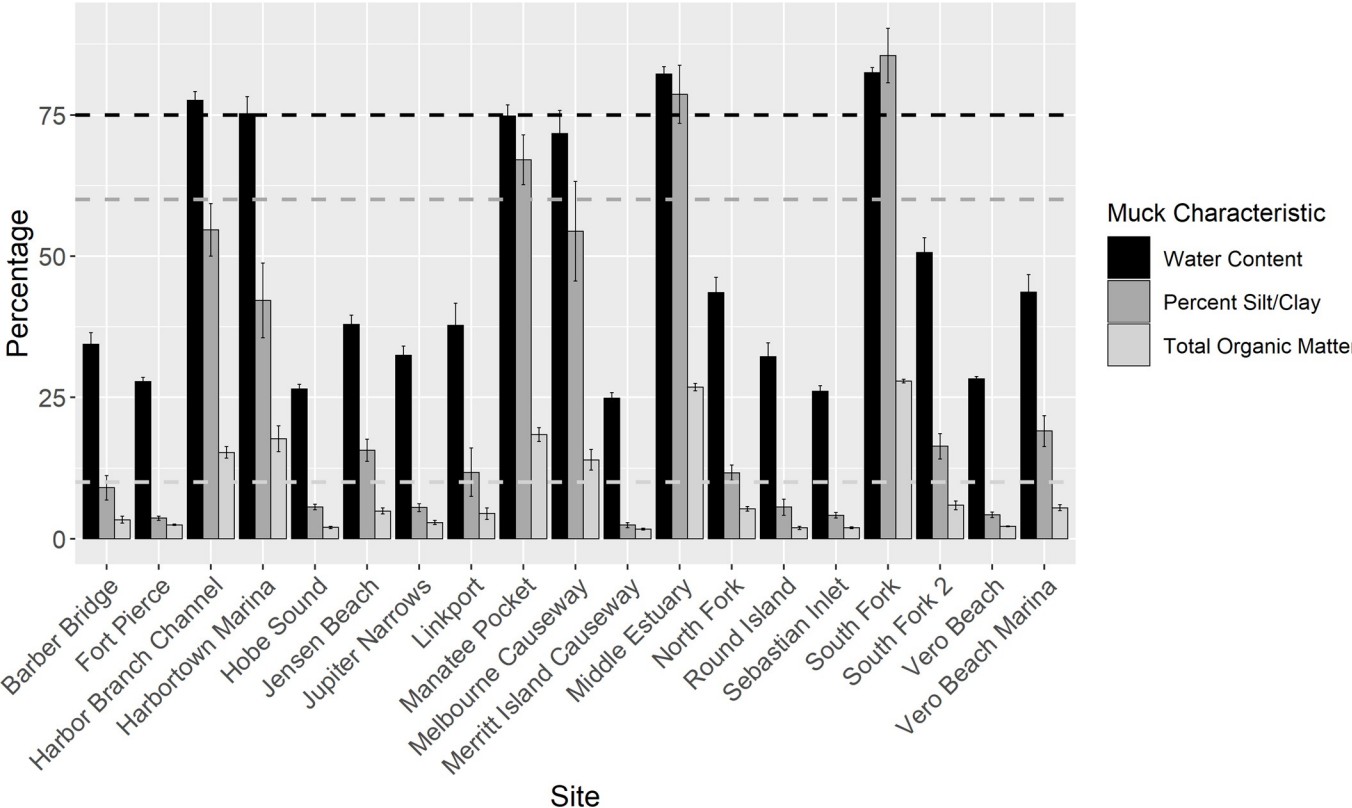

**Fig 3. Muck characteristics by site.** Bar graph, with error bars denoting standard error, summarizing the average muck characteristics associated with each site. Black bars represent water content. The black dotted line denotes the percentage water content (75% [8]) that a site, if it exceeded all three thresholds, could be considered muck. Percent silt/clay is represented by the dark gray bars with the dark gray dotted line representing the 60% [8] threshold. Total organic matter is represented by the light gray bars with the light gray dotted line representing the 10% [8] threshold.

Cu (HiHi) included HB, MP, and HT, whereas the sites with samples that had high TOM but low Cu (HiLo) included Middle Estuary, South Fork, MC, HT, Linkport, and South Fork 2. Only two samples, both from HB, were classified as having low TOM and high Cu (LoHi). The remaining 140 samples had TOM and Cu values below the thresholds (LoLo). HiHi samples (average Cu = 109 µg/g) had 3.6x and 23x more Cu than HiLo samples (30 µg/g) and LoLo samples (4.7 µg/g), respectively.

## General sequence information

There were 110 575 ASVs associated with the samples in this study. Filtering, described above, reduced the number of ASVs to 16 027. This filtering step also reduced the number of sequences from 1 857 744 to 1 598 653 (13.9%). The overall prokaryotic community had 63 phyla, 193 classes, 472 orders, 799 families, 1 315 genera, and 1 691 species.

## Alpha diversity

The mean Shannon alpha diversity was 6.45; its distribution was significantly correlated (p-values <2.2e-16) with observed ASVs (rho = 0.92), Fisher diversity (0.97), and Simpson diversity (0.69) with Spearman correlation tests. There were significant differences between Sites (BH p-value = 0.020), Estuary (0.034), Location (0.0083), Sampling Period (1.1e-15), IRL-focused Sampling Period (3.76e-15), and SLE-focused Sampling Period (3.7e-07) categories, but not by

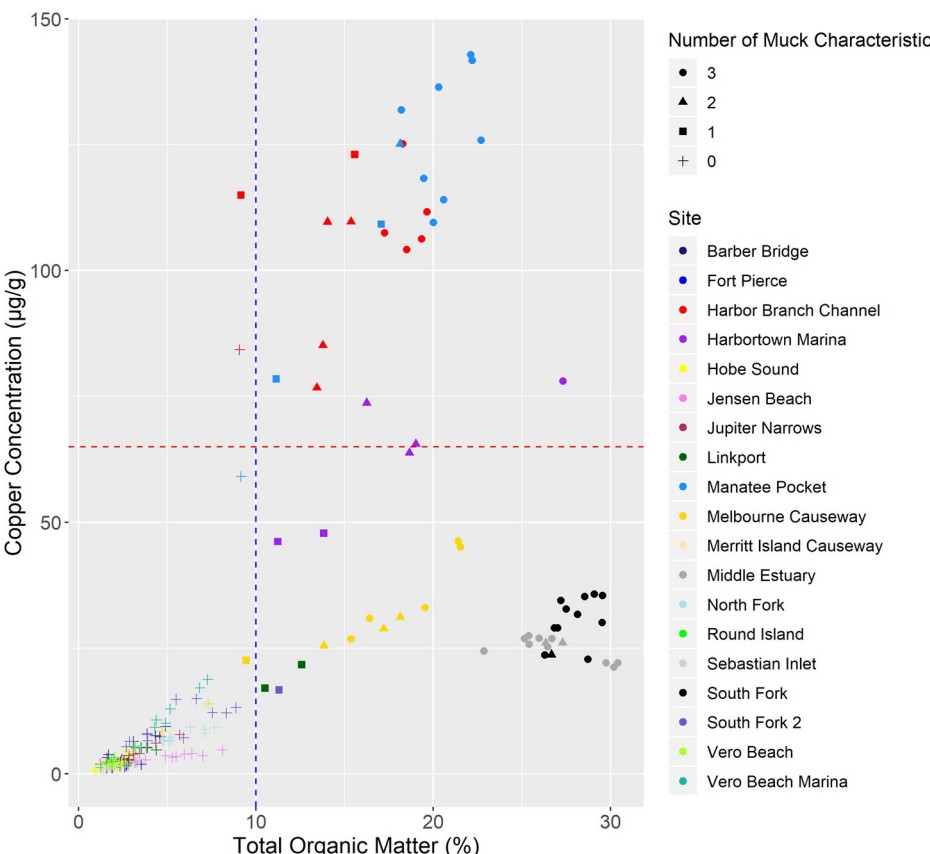

**Fig 4. Total organic matter by copper.** Point graph showing the relationship between copper (Cu) concentration (μg/g sediment) and total organic matter (TOM) percentage. Each of the 204 samples are represented by a point. Color represents the site with dark blue representing Barber Bridge, bright blue Fort Pierce, red Harbor Branch Channel, purple Harbortown Marina, yellow Hobe Sound, pink Jensen Beach, maroon Jupiter Narrows, dark green Linkport, light blue Manatee Pocket, gold Melbourne Causeway, tan Merritt Island Causeway, dark gray Middle Estuary, light blue North Fork, bright green Round Island, light gray Sebastian Inlet, black South Fork, dark purple South Fork 2, light green Vero Beach, and turquoise Vero Beach Marina. Shape represents the number of muck characteristics with circles representing three muck characteristics, triangles two, squares one, and pluses zero. The blue line at 10% [8] represents the threshold that separates the low TOM (left) from the high TOM (right) sites whereas the red line at 65 μg/g [70] separates the high Cu (above) from the low Cu (below) sites.

TOM/Cu (0.92), Muck (0.78), or Season (0.095) (S6 Table). Dunn analysis did not reveal any significantly different site pairs. Both the IRL and SLE exhibited the same patterns in terms of alpha diversity (Fig 5A). The D17 and D18 sampling periods were statistically similar to one another (IRL BH p-value = 0.16, SLE = 0.93); but were statistically dissimilar to the other two sampling periods, which were also significantly different from each other. Dunn testing at the Location level revealed that the North sites were statistically lower than the SLE, South, and South Central sites (Fig 5B).

## Prokaryotic community makeup of estuaries

The top three phyla in the IRL and SLE were the same: Proteobacteria, Bacteroidetes, and Chloroflexi (S2A Fig). The percentage of Epsilonbacteraeota was 16x higher in the IRL (2.2%) than the SLE (0.14%), whereas the percentage of Nitrospirae in the SLE (3.3%) was 8.7x more than in the IRL (0.38%). Desulfobacterales, Flavobacteriales, Anaerolineales and Steroidobacterales were four of the top five orders that overlapped between estuaries. (Fig 6A). The most common order for the SLE was Betaproteobacteriales (7.9%), which was 18x higher than the

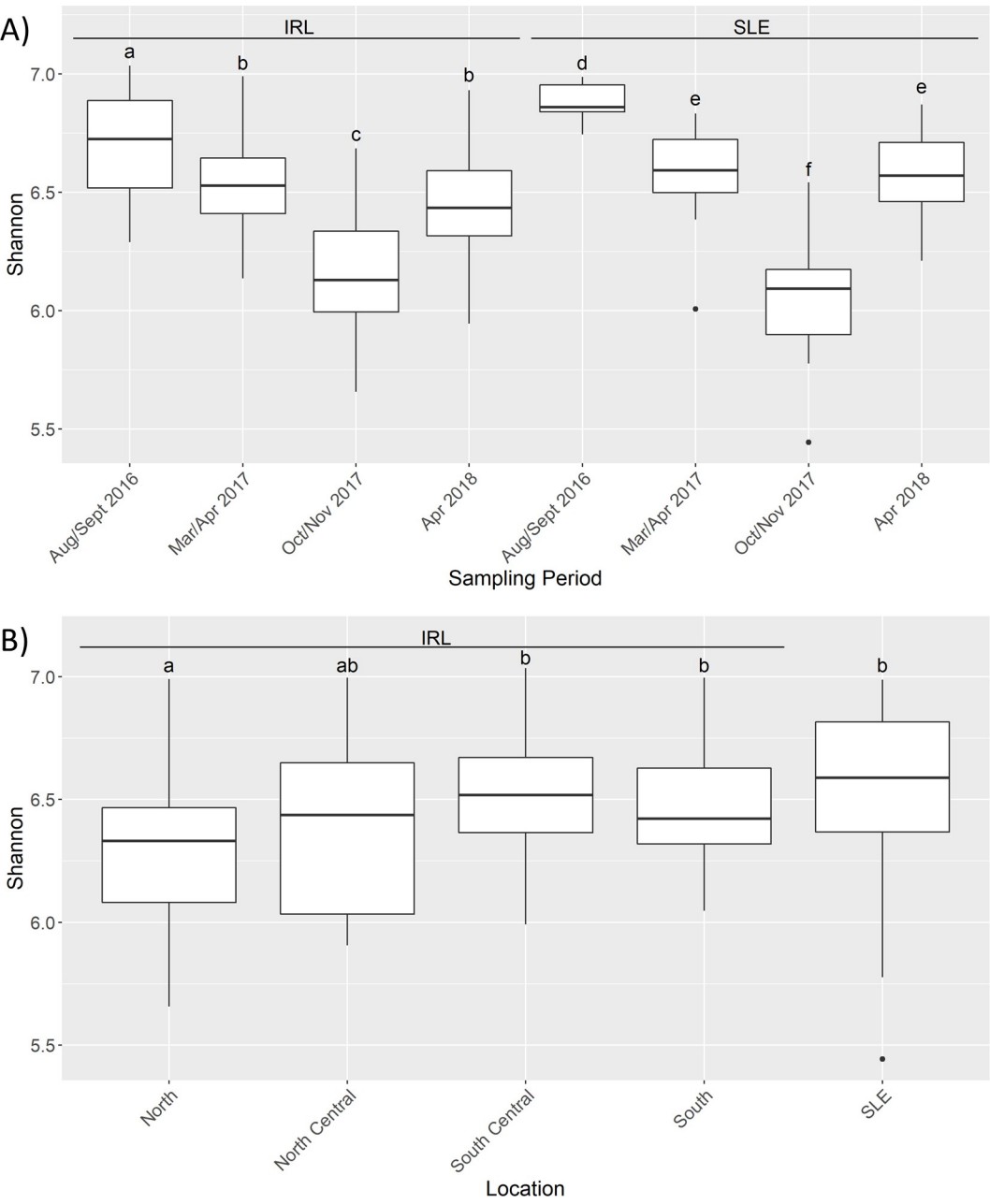

**Fig 5. Shannon diversity patterns.** Boxplots of Shannon diversity based by Estuary and Sampling Period (A) and Location (B) categories. IRL stands for Indian River Lagoon and SLE for St. Lucie Estuary. The letters on top of each boxplot denote the results from the pairwise Dunn test with different letters denoting statistical significance (Benjamini-Hochberg adjusted p values < 0.05). In A the letters show how each of the sampling periods were different within each estuary but do not denote inter-estuary comparisons. Bars denote largest and smallest values within 1.5 times the interquartile range, middle line is the median, ends of boxes are the first and third quartiles.

IRL (0.44%). The other top IRL order was Cellvibrionales (4.4%) which was 2.8x higher than in the SLE (1.5%). The following orders also occurred at levels double or greater in the IRL than in the SLE: Pirellulales (2.6x, IRL = 2.4%, SLE = 0.95%), Campylobacterales (16.0x, 2.2%, 0.14%), *B2M28* (9.6x, 1.9%, 0.20%), Actinomarinales (4.0x, 1.7%, 0.43%), and Thiotricales (3.3x, 1.7%, 0.51%). The SLE had more pronounced differences between seasons than the IRL,

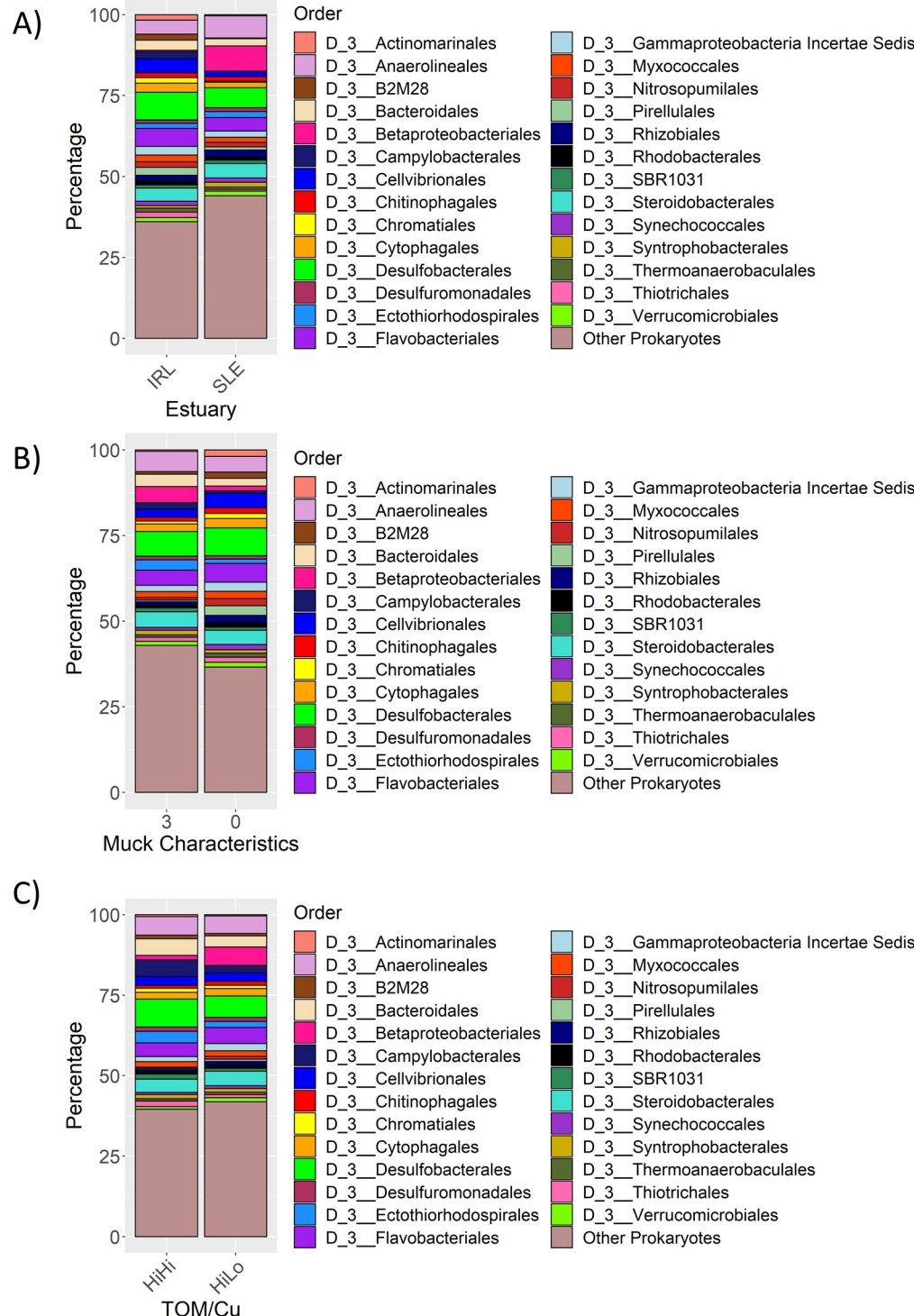

**Fig 6. Prokaryotic community patterns.** Stacked bar graphs showing the phylogenetic orders with a mean prevalence greater than 1% across all samples associated with (A) Indian River Lagoon (IRL) and St. Lucie Estuary (SLE); (B) three muck characteristics and zero muck characteristics samples and (C) high total organic matter/high copper (HiHi) and high total organic matter/low copper (HiLo) samples. TOM stands for total organic matter and Cu stands for copper.

which had the same top five orders throughout all sampling periods. (S3 Fig) In the SLE, Beta-proteobacteriales was 2.4x higher in the wet seasons (11.1%) than in the dry seasons (4.6%), whereas Flavobacteriales decreased about 4.5x between the dry (6.7%) and wet (1.5%) seasons. The following also saw decreases of at least 2x: Actinomarinales (2.1x, dry = 0.57%, wet = 0.28%), Desulfuromonadales (2.7x, 1.7%, 0.65%) Verrucomicrobiales (5.3x, 2.4%, 0.44%), Rhodobacterales (4.5x, 1.7%, 0.38%), Campylobacterales (2.8x, 0.20%, 0.072%).

## Prokaryotic community makeup of 3MC/0MC and HiHi/HiLo samples

Samples with three (3MC) and zero muck characteristics (0MC) shared four of their top five orders including Desulfobacterales, Anaerolineales, Flavobacteriales, and Steroidobacterales (Fig 6B). Bacteroidales and Cellvibrionales made up the rest of the top five for 3MC and 0MC samples respectively. Orders that were found to be at least twice as common in the 3MC samples versus the 0MC samples were: Betaproteobacteriales (3.5, 3MC = 4.8%, 0MC = 1.4%), Campylobacterales (2.3, 1.9%, 0.81%), and Ectothiorhodospirales (2.4, 3.0%, 1.3%). 0MC samples had higher levels of certain orders including Actinomarinales (4.8x, 0.39%, 1.9%), B2M28 (2.2, 0.80%, 1.8%), Nitrosopumilales (2.7, 0.76%, 2.1%), Pirellulales (5.4, 0.51%, 2.8%), Rhizobiales (2.2, 0.97%, 2.2%), and Synechococcales (2.0, 0.79%, 1.6%).

The top phyla for the high TOM and low Cu (HiLo) samples matched the order above for 3MC. High TOM and high Cu (HiHi) samples had the same top three phyla (Proteobacteria, Bacteroidetes, Chloroflexi) while Epsilonbacteraeota and Crenarchaeota replaced Acidobacteria and Planctomycetes. (S4A Fig). HiHi and HiLo shared three of the top five orders with the 3MC and 0MC samples: Desulfobacterales, Flavobacteriales, and Anaerolineales (Fig 6C). Completing the top five for HiHi was Camplyobacterales and Bacteroidales; the former was 2.2x more abundant in HiHi samples (5.2%) than in HiLo samples (2.3%). Betaproteobacteriales and Steroidobacterales completed the top five for HiLo, with the former being found 4.0x more in HiLo samples (5.8%) than in HiHi samples (1.4%). Nitrosopumilales was 2.3x higher in the HiLo (0.91%) than the HiHi samples (0.40%).

## Beta diversity

Permutational analysis of variance results showed significant differences between the IRL and SLE samples, and across samples among the Muck and TOM/Cu subcategories (Monte Carlo p-values = 0.0001) (Table 1). 3MC and 0MC samples were statistically different from one another (0.0001), and from samples with two and one muck characteristics, which were statistically similar to one another (0.76). HiHi samples were significantly dissimilar (0.0001) from LoLo and HiLo samples but not from LoHi samples (0.32). LoLo samples were also not significantly different than LoHi samples (0.072), but were from HiLo samples (0.0001). All Location combinations were significantly different than one another (0.0001) (S7 Table). All site combinations besides Barber Bridge and Vero Beach Marina (0.4) and all sampling period pairs except the SLE D17 and D18 (0.29) were statistically different from one another (<0.05).

PERMANOVAs between the samples at each site over each of the sampling periods revealed that two sites (JN and MP) did not display any significant differences between any of the sampling periods, while two others [MC and Merritt Island Causeway (MI)] were significantly different between all four sampling periods. Most of sites besides MC (0.022) and MI (0.011) were significantly similar (> 0.05) when comparing their D17 samples to D18 samples. When testing the similarity between sites within each sampling period, all site combinations in W16 were statistically different likely driving the overall statistics. Barber Bridge and Round Island, which were only sampled during W17 and D18, were statistically similar to one another during both sampling periods as well as Linkport, Sebastian Inlet, Vero Beach, and Vero Beach

**Table 1. Summarized permutational analysis of variance results.**

| Overall Parameter Category[a] | Pseudo-F | P(perms)[c] | P(MC)[d] |
|---|---|---|---|
| Pair-wise test category[b] | t statistic | | |
| **Estuary** | **42** | **0.0001** | **0.0001** |
| **TOM[e]/Cu[f]** | **7.2** | **0.0001** | **0.0001** |
| Low TOM/Low Cu, High TOM/High Cu | 2.7 | 0.0001 | 0.0001 |
| Low TOM/Low Cu, Low TOM/High Cu | 1.2 | 0.045 | 0.072 |
| Low TOM/Low Cu, High TOM/Low Cu | 3.7 | 0.0001 | 0.0001 |
| High TOM/High Cu, Low TOM/High Cu | 1.1 | 0.47 | 0.32 |
| High TOM/High Cu, High TOM/Low Cu | 2.8 | 0.0001 | 0.0001 |
| Low TOM/High Cu, High TOM/Low Cu | 1.4 | 0.034 | 0.061 |
| **Muck Characteristics** | **5.4** | **0.0001** | **0.0001** |
| 0, 2 | 1.9 | 0.0002 | 0.0001 |
| 0, 3 | 3.4 | 0.0001 | 0.0001 |
| 0, 1 | 1.6 | 0.0028 | 0.0022 |
| 2, 3 | 1.4 | 0.03 | 0.034 |
| 2, 1 | 0.82 | 0.88 | 0.76 |
| 3, 1 | 1.5 | 0.013 | 0.017 |
| **Estuary by Season** | **17** | **0.0001** | **0.0001** |
| All pairwise analyses had P(MC) values equal to 0.0001[g] | | | |
| **Location** | **10** | **0.0001** | **0.0001** |
| All pairwise analyses had P(MC) values equal to 0.0001[g] | | | |
| **Estuary by Sampling Period** | **10** | **0.0001** | **0.0001** |
| SLE-D17, SLE-D18 | 1.1 | 0.25 | 0.29 |
| All other pairwise analyses had P(MC) values less than 0.05[g] | | | |
| **Site** | **11** | **0.0001** | **0.0001** |
| Barber Bridge, Vero Beach Marina | 1 | 0.31 | 0.4 |
| All other site pairwise analysis have P(MC) values below 0.05[g] | | | |

[a]Bold text represents results from overall category and

[b]regular text represents results from the pair-wise testing results.

[c]P(perms) stands for the permutation p value

[d]P(MC) stands for Monte Carlo p value

[e]TOM stands for total organic matter, and

[f]Cu stands for copper.

[g]The full results can be seen in S7 Table.

Marina. Besides during W16, Hobe Sound was statistically similar to Jupiter Narrows and Linkport which was also statistically similar to Sebastian Inlet and Vero Beach.

## Influence of environmental variables

The environmental variable most statistically associated with prokaryotic variation between the samples was PWS (Pseudo-F = 20.396, proportion = 0.09171), followed by TOM (15.244, 0.064028), Cu (7.5017, 0.030522), and finally sediment temperature (5.5491, 0.022076) (Fig 7 and S8 Table). PWS generally decreased from the upper left corner to lower right, separating the IRL sites from the SLE sites. TOM generally decreased from the lower left corner to the upper right, separating the samples with muck characteristics from those with none. Cu generally decreased from the top of the graph, where the HiHi and LoHi samples were found, to the bottom, where the HiLo and LoLo samples were.

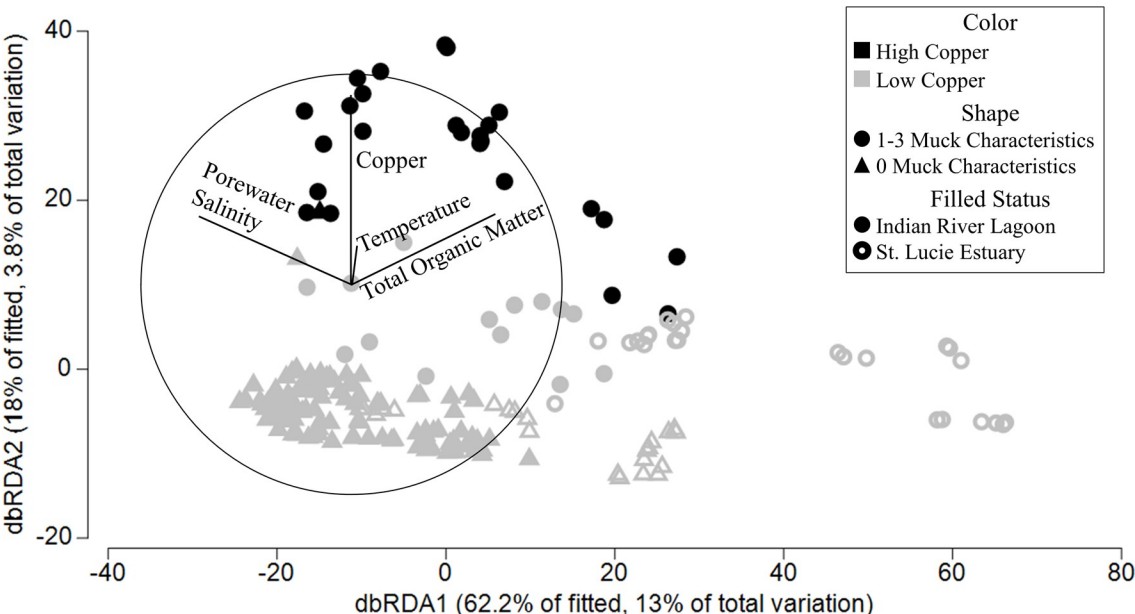

**Fig 7. Distance-based redundancy analysis.** Distance-based redundancy analysis of the sediment samples with colors representing the amount of copper [high or > 65μg/g [70] (black), low or < 65μg/g (gray)]. The shape shows the number of muck characteristics associated with a sample with circles representing 1–3 characteristics and triangles representing no muck characteristics. A filled shape is an Indian River Lagoon sample and a hollow shape represents a St. Lucie Estuary sample. The results of the distance-based linear models are shown by the lines and their associated environmental parameter and shown in S8 Table.

## Discussion

### Environmental parameters and seasonality

Sampling was scheduled to capture seasonality, but the extended impacts of Hurricane Irma which made landfall in Florida, USA on September 10, 2017 shifted one of the sampling periods from Aug-Sept 2017 to Oct-Nov 2017. PWS seasonality was more defined in the SLE than the IRL since there was a statistically lower PWS in the SLE during both wet seasons compared with the dry seasons (Fig 2A). This reflects the increased rainfall and consequent discharges from Lake Okeechobee/C44 and other canals/streams during these times (S1 Fig and S4 Table). In contrast, the PWS in the IRL was statistically different between the wet seasons (Fig 2A). Although the sediment temperature was highest during W16, W17 was either statistically similar to or lower than the two dry seasons in both the IRL and SLE (Fig 2A). In conclusion, seasonality was a prominent environmental factor in the SLE in terms of PWS but not sediment temperature; however, neither parameter showed distinguishable seasonality in the IRL. PWS generally increased from North to South in the IRL, possibly due a greater number of oceanic inlets in the south (Figs 1 and 2B) [4, 6]. Owing to higher freshwater releases, the SLE had a statistically lower PWS with a wide range (Fig 2B and S4 Table) [6].

### Muck accumulation

Muck is formed by the bacterial degradation of organic matter at the transition between freshwater and estuarine waters [7, 8]. The sites that had at least one muck sample were located near this transition in either the SLE (South Fork, Middle Estuary) or C25 (HT) (Figs 1 and 3). Samples with at least one muck characteristic were also near areas that allowed accumulation of organic matter due to restricted flow caused by the shape and bathymetry of the area (HB,

MP) or high residence time (MC) (Figs 1 and 3) [71, 72]. Although South Fork 2 is located adjacent to the C44 canal, it experiences periodic large volumes of high velocity flow that could prevent fine particles and organic matter from settling (Fig 1).

## Copper contamination

Copper can be found in the sediment near marinas due to the use of copper-antifouling paints, which explains why HB, located in a channel historically used for large boats, HT, located by a large active marina, and MP, located by a small active marina, were the only sites to have high Cu (Figs 1 and 4) [70, 73]. The site by the active Vero Beach Marina is not located in a flow-restricted area, possibly allowing the current to move contaminants away from the area (Fig 1).

## Alpha diversity patterns

Alpha diversity was highest in the warmest period (W16) and increased from north to south, matching the pattern of higher diversity in warmer environments seen in other studies (Fig 5A and 5B) [74]. The diversity drop associated with W17 samples may be related to the impact of Hurricane Irma (Fig 5A) [75].

## Estuarine prokaryotic community differences

PWS has been identified as an important factor in bacterial and archaeal community variation in other studies [64]. The most common phylum associated with either estuary was Proteobacteria, which is consistent with other estuarine studies (S2A Fig) [13, 64, 75, 76]. Proteobacteria has members that are capable of utilizing a wide variety of substrates, which allow them to occupy many different environments [77]. The most common Proteobacteria orders in the IRL and SLE included Desulfobacterales and Steroidobacterales, while the IRL had more Cellvibrionales, and the SLE had more Betaproteobacteriales (Fig 6A). Most members of Desulfobacterales, including its high-percentage families Desulfobacteraceae and Desulfobulbaceae (S2C Fig), are sulfate-reducing bacteria and have also been found in high percentages in other estuary studies [64, 78]. The family Woeseiaceae (order Steroidobacterales) (S2C Fig) has members capable of facultative sulfur- and hydrogen-based chemolithoautotrophy and is considered a core member of marine sediments [79, 80]. The family Halieaceae (order Cellvibrionales) (S2C Fig) is found in coastal marine areas and is capable of aerobic photoheterotrophic growth [81]. *Halioglobus* (S2D Fig) (Halieaceae) is capable of denitrification and requires NaCl for growth, which may be why it was more prevalent in the IRL [82]. Betaproteobacteriales was shown in multiple studies to increase in freshwater-influenced areas of estuarine lagoons, which is similar to this study where Betaproteobacteriales increased in the SLE during the wet season samples (Fig 6A and S3 Fig) [64, 76]. Flavobacteriales (phylum Bacteroidetes) (Fig 6A) is also commonly abundant in other estuary studies [75]. Most members of the family Flavobacteriaceae (S2C Fig) require NaCl or seawater salts for growth, which explains the decrease in Flavobacteriales during the wet seasons in the SLE (S3 Fig) [83]. Anaerolineaceae (phylum Chloroflexi, order Anaerolineales) (Fig 6A and S2A Fig) is comprised of obligate anaerobes with most members capable of breaking down proteinaceous carbon sources [84]. Behera et al. (2017), in another study into the effects of freshwater on a brackish lagoon, found that the phylum Acidobacteria and classes Gammaproteobacteria and Alphaproteobacteria were higher in more marine environments [64]. Our study instead found that there were more Acidobacteria in the more freshwater SLE along with relatively equal amounts of Gamma- (0.61% more in IRL) and Alphaproteobacteria (0.13% more in SLE) (S2A and S2B Fig).

Microbes that are highly abundant in a system are likely to be the actively-metabolizing part of the community; although a portion of the rare microbes can be active, they are more likely to be dormant or dead cells [85, 86]. Since Desulfobacterales and Steroidobacterales had high relative abundances, they were likely responsible for some of the sulfur cycling in the lagoon [78, 80]. Likewise, carbon cycling was likely affected by the photosynthetic Cellvibrionales and members of Anaerolineales [82, 84]. Nitrogen cycling also was affected by denitrifying members of Cellvibrionales, like Halioglobus and nitrogen-fixing members of Betaproteobacteriales [78, 81].

## Prokaryotic community shifts associated with copper and muck

Alpha diversity did not decrease in stressed sediments (sediment with muck characteristics or Cu-contamination) as seen in other studies exploring the effects of metals and clay/silt [5, 20]. While a diversity decrease can be an indicator of impaired environmental health, organisms can also become adapted to stressors with the largest drop in diversity associated with initial exposure to contamination [5]. Thus, it is possible the community has had enough time to adapt to contamination and for tolerant species to flourish [87]. There were significant differences between the 0MC and 3MC communities, which could be partially due to the smaller pore size in muck affecting the ability of some microbes to flourish [88].

TOM has also been seen as an important environmental variable in other studies [89] or studies that measured TOM-covariable parameters such as percent fines [5] or silt [22]. The top phyla matched between the 0MC and IRL samples as well as the 3MC and SLE samples (S2A and S4A Figs). This pattern likely occurred because 43.8% (21/48) of the samples in the SLE were classified as 3MC samples whereas only 12.2% (19/156) of the IRL sites were classified as 3MC samples. This could help explain why there were higher percentages of Betaproteobacteriales in the 3MC samples as well as the HiLo samples since there were also no HiHi samples in the SLE samples.

In comparison to HiLo samples and 0MC samples, HiHi and 3MC samples had higher percentages of Epsilonbacteraeota and Crenarchaeota (S4A and S5A Figs). Besaury et al. suggested that Archaea, such as the Crenarcheota, have a greater resistance to copper contamination due to their ability to sequestrate or pump out copper; this study also found Crenarchaeota flourishing in their copper-contaminated samples [24]. Members of the order Campylobacterales (phylum Epsilonbacteraeota) were also found at higher abundances in HiHi and 3MC sediments; some of its members, including the genus *Sulfurovum*, have been found in sulfide- and hydrocarbon-rich sediments similar to muck [90] as well as metal-contaminated sediments [91, 92]. (Fig 6A and 6B; S4D and S5D Figs). *Sulfurovum* is a mesophilic facultative anaerobe, requires salts for chemolithoautotrophic growth with elemental sulfur or thiosulfate as an electron donor, nitrate or oxygen as an electron acceptor, and $CO_2$ as its carbon source [93]. Conversely, 0MC and HiLo samples had higher abundances of the *Candidatus Nitrosopumilus* genus and its associated higher taxonomic ranks *Nitrosopumilus* is similar to *Sulfurovum* in that it uses $CO_2$ as its carbon source and is halophilic, but it grows chemolithoautotrophically by conducting ammonia oxidation to nitrite and is aerobic. Other families which are typically aerobic and were more abundant in the 0MC samples included Pirellulaceae and Sandaracinaceae [94–96]. This shows that the community differences between 3MC and 0MC samples were at least partially due to the former being typically more anaerobic since increased organic matter can lead to increased respiration and depletion of oxygen [8]. 3MC samples also had lower abundances of Cyanobiaceae which could be due to the increased turbidity associated with muck and its higher percentage of silt/clay [8]. Sediment microbial communities have been shown in other studies to be greatly affected by carbon sources, electron acceptors, and amount of oxygen in an area [13, 89].

## Conclusions

The most important variable causing shifts between the bacterial and archaeal communities was PWS, this was mainly due to the influence of seasonal freshwater discharges into SLE causing community differences in comparison to the IRL. Other observed differences included increases in anaerobic prokaryotes in the higher TOM 3MC samples and aerobic prokaryotes in the lower TOM 0MC samples. Tracking changes in the differentially abundant microbes present in different sediment types will allow management agencies to predict areas that are at risk of developing muck due to microbial influences or becoming sufficiently copper-contaminated to cause biological harm. This study provides the first NGS data on the bacterial and archaeal diversity of the IRL which will serve as comparison data for future IRL studies to measure the impact of anthropogenic inputs and natural disasters. This data can also be used by researchers in other estuarine areas to compare their results to determine if their systems are facing similar shifts in the prokaryotic communities due to similar anthropogenic impacts.

Future studies should be performed with greater sequencing depth and higher sampling frequency, which could allow more of the diversity and rarer taxa in the samples to be captured. Future studies should also take more samples at site during each sampling period to further delineate site to site differences with higher statistical power. One of the main limitations of 16S is that it can only be used to predict functional differences between samples, thus shotgun metagenomics should be used in future studies to identify functional differences between sites or sample types. Incorporating the measurement of anoxia and biogeochemical cycles would help to further delineate which environmental variables are causing shifts to the bacterial and archaeal communities between sediment types and geographical locations.

## Supporting information

**S1 Fig. National Weather Service temperature and rain sum patterns.** National Weather Service NOWData showing the four sampling periods (Aug-Sep 2016 (dark orange), Mar-Apr 2017a (dark green), Oct-Nov 2017b (light orange), Apr 2018 (light green)) as well as the historical temperature (A) and rain sum (B) during those months for the years 1990–2018 (dark gray). Bars denote standard error.
(TIF)

**S2 Fig. Other taxonomic levels by Estuary.** Stacked bar graphs showing the phyla (A), classes (B), orders (C), families (D), and genera (E) that have a mean of greater than 1% across all samples. These graphs show the differences between the two main basins of the study, (Indian River Lagoon (IRL) or St. Lucie Estuary (SLE)).
(TIF)

**S3 Fig. Estuary orders by sampling period.** Stacked bar graph showing the orders with a mean greater than 1% across all samples grouped by estuary (Indian River Lagoon (IRL) or St. Lucie Estuary (SLE)) and sampling period (Aug/Sept 2016, Mar/Apr 2017, Oct/Nov 2017, and Apr 2018).
(TIF)

**S4 Fig. Other taxonomic levels by muck classification.** Stacked bar graphs showing the phyla (A), classes (B), orders (C), families (D), and genera (E) that have a mean of greater than 1% across all samples. These graphs show the differences between the samples with three and zero muck characteristics.
(TIF)

**S5 Fig. Other taxonomic levels by total organic matter/copper classification.** Other taxonomic levels by Total Organic Matter/Copper classification. Stacked bar graphs showing the phyla (A), classes (B), orders (C), families (D), and genera (E) that have a mean of greater than 1% across all samples. These graphs show the differences between the samples with high TOM and copper (HiHi) and high TOM and low copper (HiLo).
(TIF)

**S1 Table. GPS coordinates.** [a]NWS stands for National Weather Service [30], [b]IRFWCD for Indian River Farms Water Control District, [c]USGS for United States Geological Service [32], [d]SFWMD for South Florida Water Management District [31].
(DOCX)

**S2 Table. Measured environmental variables per sample.** Sediment temperature was determined using a thermometer, water content by weight loss during oven drying, total organic matter by weight loss in a muffle furnace, grain size fractions (gravel, sand, and silt/clay) were determined using wet sieving, and copper (Cu) and iron (Fe) were measured using an atomic adsorption spectrometer. See manuscript for details.
(DOCX)

**S3 Table. Metadata per sample.** Table containing information pertaining to each sample such as when (Sampling Season, Season) and where (Site, Location, Estuary) it was taken. A sample was considered to have a muck characteristic if it exceeded one of the muck thresholds: 10% for total organic matter, 60% for silt/clay fraction, and 75% for water content [8]. Additionally, a sample was considered to have high copper if it exceeded 65 μg/g [70]. [a]IRL for Indian River Lagoon, [b]SLE for St. Lucie Estuary, [c]TOM stands for total organic matter, [d]Cu for copper, [e]LoLo for low TOM/low Cu, [f]HiHi for high TOM/low Cu, [g]LoHi for low TOM/low Cu, and [h]HiLo stands for high TOM/low Cu.
(DOCX)

**S4 Table. Average monthly means for canal daily discharges (ft$^3$/s).** The average streamflow from all fourteen stream/canals separately (A), averaged together regionally (B), and all together (C) during the months before and during each sampling period. Data was taken from the [*]United States Geological Services online database [32] or the [**]South Florida Water Management District's DBHYDRO online database [31]. [a]The regional location each canal/stream was found in, [b]data from the month before and months during each sampling period and [c]the entire survey. [d]IRL stands for Indian River Lagoon.
(DOCX)

**S5 Table. Environmental parameters statistical analysis.** [a]Bold text is associated with testing the overall differences within a category with Kruskal-Wallis. [b]Regular text is associated with pair-wise Dunn testing. [c]BH stands for Benjamini-Hochberg, [d]IRL stands for Indian River Lagoon and [e]SLE stands for St. Lucie Estuary.
(DOCX)

**S6 Table. Shannon diversity statistical analysis.** [a]Bold text is associated with testing the overall differences within a category with Kruskal-Wallis or Mann-Whitney use with a [*] indicating the latter was used. [b]Regular text is associated with pair-wise Dunn testing. [c]BH stands for Benjami-Hochberg, [d]TOM for total organic matter, [e]Cu for copper, [f]IRL for Indian River Lagoon and [g]SLE for St. Lucie Estuary.
(DOCX)

**S7 Table. Full permutational analysis of variance results.** [a]Bold text is associated with testing the overall differences within a category and [b]regular text is associated with pair-wise testing. [c]P(perms) stands for permutational p value, P(MC) for Monte-Carlo p value, [e]TOM for total organic matter, [f]Cu for copper, [g]IRL stands for Indian River Lagoon and [h]SLE for St. Lucie Estuary.
(DOCX)

**S8 Table. Distance-based linear model results.** [a]Marginal statistical tests are displayed for all variables and [b]sequential statistical results are shown only if the variable was determined to contribute a statistically significant amount of variation between microbial samples (p value < 0.05). [c]SS for sum of squares and [d]AICc stands for An Information Criterion.
(DOCX)

# Acknowledgments

We thank Dennis Hanisak and Joshua Voss for their guidance throughout this project; Gabrielle Barbarite, Austin Fox, Stacey Fox, Dedra Harmody, John Hart, Hunter Hines, Brandon McHenry, and Carlie Perricone for their advice and contributions to the research. Fig 1 was created by Kristen Davis. A special thanks to Emily Sniegowski for her support and thoughtful edits.

# Author Contributions

**Conceptualization:** David J. Bradshaw, II, John H. Trefry, Peter J. McCarthy.

**Data curation:** David J. Bradshaw, II, Nicholas J. Dickens, John H. Trefry, Peter J. McCarthy.

**Formal analysis:** David J. Bradshaw, II, Nicholas J. Dickens, John H. Trefry, Peter J. McCarthy.

**Funding acquisition:** David J. Bradshaw, II, Nicholas J. Dickens, Peter J. McCarthy.

**Investigation:** David J. Bradshaw, II, Nicholas J. Dickens, John H. Trefry, Peter J. McCarthy.

**Methodology:** David J. Bradshaw, II, Nicholas J. Dickens, John H. Trefry, Peter J. McCarthy.

**Project administration:** David J. Bradshaw, II, Peter J. McCarthy.

**Resources:** David J. Bradshaw, II, Nicholas J. Dickens, John H. Trefry, Peter J. McCarthy.

**Software:** David J. Bradshaw, II, Nicholas J. Dickens.

**Supervision:** Nicholas J. Dickens, John H. Trefry, Peter J. McCarthy.

**Validation:** David J. Bradshaw, II, Nicholas J. Dickens, Peter J. McCarthy.

**Visualization:** David J. Bradshaw, II, Peter J. McCarthy.

**Writing – original draft:** David J. Bradshaw, II.

**Writing – review & editing:** David J. Bradshaw, II, Nicholas J. Dickens, John H. Trefry, Peter J. McCarthy.

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
