## [Decision Letter · Decision Letter 0]

4 Aug 2020

PONE-D-20-20716

Defining the sediment microbiome of the Indian River Lagoon, FL, USA, an Estuary of National Significance

PLOS ONE

Dear Dr. Bradshaw,

Thank you for submitting your manuscript to PLOS ONE. After careful consideration, we feel that it has merit but does not fully meet PLOS ONE’s publication criteria as it currently stands. Therefore, we invite you to submit a revised version of the manuscript that addresses the points raised during the review process.

We look forward to receiving your revised manuscript.

Kind regards,

Hans-Uwe Dahms, Ph.D.

Academic Editor

PLOS ONE

Additional Editor Comments:

The reporting is straight-forward and clear. The approach of grouping of results from the individual samples based on the study locations (IRL and SLE), however, needs reconsideration. These mask the variations between the samples. Also, the sampling strategy needs to be elaborated regarding the number of samples collected at each of the sites and their frequencies during the sampling periods. The quality of the figures is too low. Other comments are given in the marked manuscript

sampling strategies need more detail regarding the number, sites, and frequencies of the collected samples throughout the sampling periods. Contrast of figures need to be enhanced. Existing REFERENCES need to be updated. The ecological significance of the obtained results need to be discussed particular with respect to functional differences.

Journal Requirements:

3.Thank you for stating the following in the Financial Disclosure section:

"This research was supported by the following organizations and agencies: the Harbor Branch Oceanographic Institute Foundation (https://hboifoundation.org/) (DJB), the Everglades Foundation (https://evergladesfoundation.org/) (DJB), and the Save Our Seas License Plate (https://myfloridaspecialtyplate.com/save-our-seas.html) (PJM).The funders had no role in study design, data collection and analysis, decision to publish, or preparation of the manuscript."

We note that you received funding from a commercial source: MyFloridaSpecialtyPlate

We note that one or more of the authors are employed by a commercial company: Amazon.com, Inc..

4.1. Please provide an amended Funding Statement declaring this commercial affiliation, as well as a statement regarding the Role of Funders in your study. If the funding organization did not play a role in the study design, data collection and analysis, decision to publish, or preparation of the manuscript and only provided financial support in the form of authors' salaries and/or research materials, please review your statements relating to the author contributions, and ensure you have specifically and accurately indicated the role(s) that these authors had in your study. You can update author roles in the Author Contributions section of the online submission form.

4.2. Please also provide an updated Competing Interests Statement declaring this commercial affiliation along with any other relevant declarations relating to employment, consultancy, patents, products in development, or marketed products, etc. 

5. We note that 1 in your submission contain map images which may be copyrighted. All PLOS content is published under the Creative Commons Attribution License (CC BY 4.0), which means that the manuscript, images, and Supporting Information files will be freely available online, and any third party is permitted to access, download, copy, distribute, and use these materials in any way, even commercially, with proper attribution. For these reasons, we cannot publish previously copyrighted maps or satellite images created using proprietary data, such as Google software (Google Maps, Street View, and Earth). For more information, see our copyright guidelines: http://journals.plos.org/plosone/s/licenses-and-copyright.

5.1.    You may seek permission from the original copyright holder of Figure 1 to publish the content specifically under the CC BY 4.0 license.

5.2.    If you are unable to obtain permission from the original copyright holder to publish these figures under the CC BY 4.0 license or if the copyright holder’s requirements are incompatible with the CC BY 4.0 license, please either i) remove the figure or ii) supply a replacement figure that complies with the CC BY 4.0 license. Please check copyright information on all replacement figures and update the figure caption with source information. If applicable, please specify in the figure caption text when a figure is similar but not identical to the original image and is therefore for illustrative purposes only.

Reviewers' comments:

Reviewer's Responses to Questions

**Comments to the Author**

1. Is the manuscript technically sound, and do the data support the conclusions?

Reviewer #1: Partly

Reviewer #2: Yes

2. Has the statistical analysis been performed appropriately and rigorously? 

Reviewer #1: Yes

Reviewer #2: Yes

3. Have the authors made all data underlying the findings in their manuscript fully available?

Reviewer #1: Yes

Reviewer #2: Yes

4. Is the manuscript presented in an intelligible fashion and written in standard English?

Reviewer #1: Yes

Reviewer #2: Yes

5. Review Comments to the Author

Reviewer #1: This manuscript describes the diversity of bacteria and archaea at the Indian River Lagoon and the St. Lucie Estuary using a metabarcoding approach over a two-year period and also compares polluted and non-polluted sites. The reporting is straight-forward and clear. However, I do not agree with the approach of grouping of results from the individual samples based on the study locations (IRL and SLE). This would have masked the variations between the samples. Also, the sampling strategy needs to be elaborated regarding the number of samples collected at each of the sites and their frequencies during the sampling periods. The quality of the figures is too low. Other comments are given in the marked manuscript.

Reviewer #2: This contribution discusses original findings from the first study of the sediment microbiome of the Indian River Lagoon (IRL) which has high biodiversity and economic value, but little is known about its microbial communities. The research claimed to fill this knowledge gap and to assess the spatiotemporal changes of microbial populations due to stressors. These stressors, that many other estuaries also face worldwide, include freshwater discharges, and development of fine-grained, organic-rich, anoxic sediment This study was conducted over two years during four sampling periods along a 100-mile stretch of the IRL. 16S rRNA amplicon sequencing and analysis was performed on 204 sediment samples. The most common IRL microbes were consistent with other studies into subtropical, estuarine microbiomes including the dominance of Proteobacteria. Many of the highly abundant orders in this phylum had members capable of contributing to biogeochemical cycles. The main environmental influences on microbial variation were claimed to be porewater salinity, total organic matter (TOM) and copper (Cu). Anaerobic prokaryotes were more associated with high TOM and Cu samples whereas aerobic prokaryotes were more associated with lower TOM and Cu samples. This study provides a novel baseline to assess the potential impact of changes due to anthropogenic factors, natural disasters, and management decisions on the distribution sediment microbial populations of the ecologically important IRL. The most important variable causing shifts between the microbiomes was PWS, this was mainly 500 due to the influence of seasonal freshwater discharges into SLE causing microbiome differences of the sediment microbiome of the Indian River Lagoon in comparison to the IRL.

The sampling strategies need more detail regarding the number, sites, and frequencies of the collected samples throughout the sampling periods. Contrast of figures need to be enhanced. Existing REFERENCES need to be updated. The ecological significance of the obtained results need to be discussed particular with respect to functional differences. END

6. PLOS authors have the option to publish the peer review history of their article (what does this mean?). If published, this will include your full peer review and any attached files.

Reviewer #1: No

Reviewer #2: No

---

## [Author Response · Author response to Decision Letter 0]

18 Sep 2020

We would like to thank Dr. Hans-Uwe Dahms and the reviewers for their time and comments. Please find below our response to each of your comments. Thank you again for the opportunity to revise our manuscript with your helpful edits. 

Reviewer comments made on pdf version of manuscript

Title/Authors/Affiliations Page

1. Comment while highlighting “microbiome” in Line 1: “Is it better to use prokaryotes or archaea/bacteria throughout the ms? You did not look into fungal, viral, protozoan diversities.”

The reviewer correctly indicated that this is not a full microbiome study, thus all instances of “microbiome” or “microbial communities” that are in reference to just this study were changed to prokaryotes or bacterial/archaeal communities starting with the title (Starting with Line 4 of Revised Manuscript With Track Changes). 

Abstract

2. Edit of removing “a” from Line 19; corrected Line 22

3. Comment while highlighting “higher” in Lines 27-28: “higher in terms of abundance? or species richness?”

Higher Betaproteobacteriales and Flavobacteriales were clarified to indicate that this referred to their relative abundances (Lines 31-32).

4. Comment while highlighting “freshwater” in Line 27: “exclusively?”

Thank you for indicating this needed clarification. Betaproteobacteriales is not exclusively a freshwater-associated bacterial group but has been shown in other studies to have a higher relative abundance in freshwater-associated regions of estuaries [1,2]. The words “a freshwater associated organism” were removed from the Abstract (Lines 31-32). This statement is more appropriate in just the Discussion section of the document where it can be further explained (Lines 483-484). 

5. Reviewer crossed out “organism” in Line 28 replacing with “bacterial group”, “organisms” in Line 30 replacing with “prokaryotes” and highlighted “organism” in Line 32. 

Thank you for providing alternative verbiage for the use of organism(s). References to organism(s) were changed to bacterial group(s) or prokaryote(s) throughout the document to reflect a more appropriate word choice starting with the Abstract (Starting with Line 26). 

Introduction

6. Comment while highlighting sentence “This study…TOM samples.” in Lines 78-82: “Can be more specific as you only looked into bacterial/archaeal diversities.”

Please see response to Reviewer Comment #1 above. 

7. Comment while highlighting “Aug-Sep 2016 (W16), Mar-Apr 2017 (D17), Oct-Nov 2017(W17), and Apr 2018 (D18).” in Lines 115-116: “Can these be more specific? Were equal numbers of samples taken during each of the periods? Are the samples representative of the sites/seasons?”

Thank you for indicating to us that our sampling strategy was unclear. The “Site selection” (Lines 96-108) and “Sample collection” (Lines 129-135) sections have been updated to make it more clear that 15 sites were sampled during the first two sampling periods and 19 sites during the second two sampling periods in triplicate to get representative samples of those sites and sampling periods.

8. Reviewer highlighted M� in Line 131. 

This statement was updated to indicate that ultrapure water was used instead of defining the water used by its resistance (M�) (Line 150 and 159). The authors were unsure why this was highlighted and hope that this addresses the reviewer’s concern. 

Results

9. Comment while highlighting paragraph starting with Measured air temperatures…South Fork 2 site (S4 Table)” in Lines 199-204: “How can the data be treated as baseline data for future studies when the air temperatures and rainfall were abnormal during the sample years?

As the reviewer correctly pointed out, this study will act as comparison data for future studies instead of as a baseline due to the abnormal environmental conditions during the sampling periods. The manuscript was updated to reflect this in the Abstract (Lines 39-40) and Conclusions (Lines 556). 

10. Edit add “.” to end of sentence in Line 278; corrected Line 308

11. Comment while highlighting subtitle “Microbial community makeup of estuaries” in Line 302: “Individual samples should be compared to examine variations of the prokaryotic diversities at the sample level. How variable were the diversities between samples collected at the different sampling sites during the different sampling months?”

The authors appreciate this comment and agree that that the IRL versus SLE comparison will mask changes at the sample level. But, the comparison between the IRL and SLE samples is still important to explore due to the differences in freshwater discharge rates between the regions during the study and the distinct separation of samples in these regions in the PCoA (Fig 6). To further stress this point we added a paragraph outlining the differences in freshwater discharges between regions and sampling periods (Lines 227-234) and added another table (S4C Table) to the S4 Table group stressing the differences between the IRL and SLE. 

To complement this regional comparison, we also grouped similar sample types together despite their regional and sampling period differences to focus on ecologically important anthropogenic impacts of muck and copper accumulation. This also allowed us to increase statistical power. This was clarified in Lines 101-104. 

The reviewer is correct in that comparing samples at the site level over time or between sites during one sampling season is important. But due to our study design and alternative goals this is a low statistical power comparison since there were only three samples at each site during each sampling period. But to address this concern we conducted PERMANOVA analysis at each site between sampling seasons and during each sampling period between sites and added the results to S7 Table. The major trends are briefly discussed in Lines 387-402. 

12. Edit “make” to “made” in Line 332; corrected Line 362

Reviewer comments summarized in email

13. Reviewer #1: The reporting is straight-forward and clear. However, I do not agree with the approach of grouping of results from the individual samples based on the study locations (IRL and SLE). This would have masked the variations between the samples.

Please see response to Comment #11 above.

14. Reviewer #1: Also, the sampling strategy needs to be elaborated regarding the number of samples collected at each of the sites and their frequencies during the sampling periods.

15. Reviewer #2: The sampling strategies need more detail regarding the number, sites, and frequencies of the collected samples throughout the sampling periods. 

Please see response to Comment #7 above.

16. Reviewer #1: The quality of the figures is too low. 

17. Reviewer #2: Contrast of figures need to be enhanced.

PACE was utilized to determine that the figures did actually meet PLOS requirements since PACE did not make any corrections. PACE was used to rename the files according to PLOS ONE standards. 

In reference to the above reviewer’s comments, the authors believe that the image quality was affected by being converted to the compiled submission PDF. This is supported by the following statement on PLOS ONE’s submission guidelines page (https://journals.plos.org/plosone/s/submission-guidelines) “The compiled submission PDF includes low-resolution preview images of the figures after the reference list. The function of these previews is to allow you to download the entire submission as quickly as possible. Click the link at the top of each preview page to download a high-resolution version of each figure. Links to download Supporting Information files are also available after the reference list.”

19. Reviewer #2: Existing REFERENCES need to be updated.

The authors thank the reviewer for indicating that we needed to update our references. To accomplish this, all websites provided were checked to make sure they still worked, and were updated if they did not, and all “cited …” statements were updated to 6 Sep 2020 to reflect this. We used the citation manager Mendeley with the PLOS ONE style they had on file. One citation was removed since it was the preprint for QIIME2 not the published Nature Biotechnology article.

20. Reviewer #2: The ecological significance of the obtained results need to be discussed particular with respect to functional differences.

The authors agree that understanding the functional differences between prokaryotic communities is important but, since this is a 16S study, that is outside the scope of our data. We did not feel comfortable making conclusions regarding functional differences between samples and sample types unless the trait in question can be attributed to all or most of the members of a bacterial group. We covered this in Lines 497-504 where we discussed the major orders and their possible contributions to the nutrient cycling in the sediment. To address this limitation, we expanded our conclusions section to stress the importance of future shotgun sequencing studies Lines 563-566. 

Notes from email:

Thank you for reminding us to double-check our formatting, the authors are sorry that they missed some style requirements and the Title, Authors, and Affiliations page was edited to reflect these requirements. Commas were switched to after the author labels (Line 7) and the current address for Nicholas J. Dickens was removed (Line 13) to reflect that his contributions to this work only occurred during his time at Harbor Branch Oceanographic Institute. Based upon careful review of the main body style requirements, no edits were made since the document already reflected those requirements. All file names were changed to directly match what was stated in email, guidance from website, and using PACE. 

22. In your Methods section, please provide additional information regarding the permits you obtained for the work. Please ensure you have included the full name of the authority that approved the field site access and, if no permits were required, a brief statement explaining why.

The authors appreciate you pointing out this missing statement. No permits were necessary and a statement reflecting this is included in lines 129-130.

23. Thank you for stating the following in the Financial Disclosure section: "This research was supported by the following organizations and agencies: the Harbor Branch Oceanographic Institute Foundation (https://hboifoundation.org/) (DJB), the Everglades Foundation (https://evergladesfoundation.org/) (DJB), and the Save Our Seas License Plate (https://myfloridaspecialtyplate.com/save-our-seas.html) (PJM).The funders had no role in study design, data collection and analysis, decision to publish, or preparation of the manuscript." We note that you received funding from a commercial source: MyFloridaSpecialtyPlate Please provide an amended Competing Interests Statement that explicitly states this commercial funder, along with any other relevant declarations relating to employment, consultancy, patents, products in development, marketed products, etc.

Thank you for informing us about this discrepancy, the Competing Interests Statement (below) has been amended to address the above issue. 

The authors have declared that no competing interests exits. In specific relation to the commercial funder MyFloridaSpecialtyPlate, none of the authors have any of the following financial competing interests to declare: ownership of stocks or shares, paid employment or consultancy, board membership, patent applications, or marketed products. None of the funders involved in this research have any policies nor do any of the authors have an involvement with these funders which would alter our adherence to PLOS ONE policies on sharing data and materials. 

24. Thank you for stating the following in the Competing Interests section:"The authors have declared that no competing interests exist." We note that one or more of the authors are employed by a commercial company: Amazon.com, Inc..

4.1. Please provide an amended Funding Statement declaring this commercial affiliation, as well as a statement regarding the Role of Funders in your study. If the funding organization did not play a role in the study design, data collection and analysis, decision to publish, or preparation of the manuscript and only provided financial support in the form of authors' salaries and/or research materials, please review your statements relating to the author contributions, and ensure you have specifically and accurately indicated the role(s) that these authors had in your study. You can update author roles in the Author Contributions section of the online submission form.

Please also include the following statement within your amended Funding Statement.“The funder provided support in the form of salaries for authors [insert relevant initials], but did not have any additional role in the study design, data collection and analysis, decision to publish, or preparation of the manuscript. The specific roles of these authors are articulated in the ‘author contributions’ section.”

4.2. Please also provide an updated Competing Interests Statement declaring this commercial affiliation along with any other relevant declarations relating to employment, consultancy, patents, products in development, or marketed products, etc. 

Thank you for pointing this out, as stated above, the current address for Nicholas J. Dickens was removed to reflect that his involvement in this study only occurred while he was exclusively a member of the Harbor Branch Oceanographic Institute faculty before his present employment by Amazon.com Inc. He is still affiliate faculty at HBOI and his email address in this role is dickensn@fau.edu. 

The authors believe that the above adjustment to the Funding and Competing Interest Statement is now no longer needed due to this change and clarification. In relation to Dr. Dickens Author Contributions Role as a funder, his startup funds provided by the Harbor Branch Oceanographic Institute Foundation was used to fund the Research Assistantship of David J. Bradshaw during Fall 2018, Spring 2019, Fall 2019, and Spring 2020. The Funding Statement (below) has been updated to reflect this. 

This research was supported by the following organizations and agencies: the Harbor Branch Oceanographic Institute Foundation (https://hboifoundation.org/) (DJB, NJD), the Everglades Foundation (https://evergladesfoundation.org/) (DJB), and the Save Our Seas License Plate (https://myfloridaspecialtyplate.com/save-our-seas.html) (PJM).The funders had no role in study design, data collection and analysis, decision to publish, or preparation of the manuscript.

25. We note that 1 in your submission contain map images which may be copyrighted. All PLOS content is published under the Creative Commons Attribution License (CC BY 4.0), which means that the manuscript, images, and Supporting Information files will be freely available online, and any third party is permitted to access, download, copy, distribute, and use these materials in any way, even commercially, with proper attribution. For these reasons, we cannot publish previously copyrighted maps or satellite images created using proprietary data, such as Google software (Google Maps, Street View, and Earth). For more information, see our copyright guidelines: http://journals.plos.org/plosone/s/licenses-and-copyright.

Fig 1 was recreated using a public domain source for the image (Florida Fish and Wildlife Conservation Commission), thus it is not copyrighted. The content permission statement is thus not needed, but the figure legend has been updated to state the source of the image (Line 123-124).

---

## [Editor Report · Decision Letter 1]

8 Oct 2020

Defining the sediment prokaryotic communities of the Indian River Lagoon, FL, USA, an Estuary of National Significance

PONE-D-20-20716R1

Dear Dr. Bradshaw,

We’re pleased to inform you that your manuscript has been judged scientifically suitable for publication and will be formally accepted for publication once it meets all outstanding technical requirements.

Kind regards,

Hans-Uwe Dahms, Ph.D.

Academic Editor

PLOS ONE

Additional Editor Comments (optional):

The 1st revised version of the MS has reached a standard that is acceptable for publication.
---

## [Editor Report · Acceptance letter]

16 Oct 2020

PONE-D-20-20716R1 

Defining the sediment prokaryotic communities of the Indian River Lagoon, FL, USA, an Estuary of National Significance 

Dear Dr. Bradshaw II:

I'm pleased to inform you that your manuscript has been deemed suitable for publication in PLOS ONE. Congratulations! Your manuscript is now with our production department. 

Kind regards, 

on behalf of

Dr. Hans-Uwe Dahms 

Academic Editor

PLOS ONE